# Near-Infrared Photoresponse in Ge/Si Quantum Dots Enhanced by Photon-Trapping Hole Arrays

**DOI:** 10.3390/nano11092302

**Published:** 2021-09-04

**Authors:** Andrew I. Yakimov, Victor V. Kirienko, Aleksei A. Bloshkin, Dmitrii E. Utkin, Anatoly V. Dvurechenskii

**Affiliations:** 1Rzhanov Institute of Semiconductor Physics, Siberian Branch of the Russian Academy of Science, 630090 Novosibirsk, Russia; victor@isp.nsc.ru (V.V.K.); bloshkin@isp.nsc.ru (A.A.B.); utkinde@isp.nsc.ru (D.E.U.); dvurech@isp.nsc.ru (A.V.D.); 2Physical Department, Novosibirsk State University, 630090 Novosibirsk, Russia

**Keywords:** quantum dots, near-infrared photodetection, photon-trapping nanostructures, telecom

## Abstract

Group-IV photonic devices that contain Si and Ge are very attractive due to their compatibility with integrated silicon photonics platforms. Despite the recent progress in fabrication of Ge/Si quantum dot (QD) photodetectors, their low quantum efficiency still remains a major challenge and different approaches to improve the QD photoresponse are under investigation. In this paper, we report on the fabrication and optical characterization of Ge/Si QD pin photodiodes integrated with photon-trapping microstructures for near-infrared photodetection. The photon traps represent vertical holes having 2D periodicity with a feature size of about 1 μm on the diode surface, which significantly increase the normal incidence light absorption of Ge/Si QDs due to generation of lateral optical modes in the wide telecommunication wavelength range. For a hole array periodicity of 1700 nm and hole diameter of 1130 nm, the responsivity of the photon-trapping device is found to be enhanced by about 25 times at λ=1.2 μm and by 34 times at λ≈1.6 μm relative to a bare detector without holes. These results make the micro/nanohole Ge/Si QD photodiodes promising to cover the operation wavelength range from the telecom O-band (1260–1360 nm) up to the L-band (1565–1625 nm).

## 1. Introduction

Ge/Si quantum dot (QD) infrared (IR) photodetectors (QDIPs) can operate in the near-IR telecommunication wavelength range (1.2–1.6 μm). The main advantage of such structures is that they can be fabricated on Si substrates; consequently, very large matrices can be fabricated by monolithic integration with silicon field-effect transistors and fast signal processing electronics. However, the currently reached sensitivity to IR radiation is low. The main reason is a low density of states associated with QDs, and thereby a low absorption coefficient. To increase the absorption and hence the efficiency of QDIPs it is therefore necessary to introduce photon traps. Several studies have been dedicated to the problem of a low photoconversion efficiency in QDIPs, focusing mainly on the plasmonic effects in metallic grating. It has recently been found that the integration of Ge/Si heterostructures containing layers of QDs with regular subwavelength gratings of holes in metallic films or with arrays of metallic nanoparticles on the surface of a semiconductor increases the responsivity in the mid-IR (3–5 μm) [1] and near-IR [2] ranges by an order of magnitude. These results were explained by the excitation of localized surface plasmon modes or plasmon polariton waves effectively interacting with dipole transitions in QDs. Similar results were also obtained for longwave (8–12 μm) detectors based on InAs/GaAs QDs [3,4,5,6]. Disadvantages of metallic metasurfaces that allow conversion of the incident electromagnetic radiation into the surface plasmons are high ohmic losses in a metal [7] and a small penetration depth of the plasmon field to a semiconductor, particularly for short wavelengths.

In the meantime, there is an increasing interest in resonant all-dielectric elements, as they offer a low-loss alternative to plasmon structures [8,9,10,11,12,13]. The proposal of using a pure dielectric light-trapping structure to enhance the absorption or emission of optoelectronics devices such as LEDs [14,15,16,17,18,19] or solar cells [20,21,22,23,24,25,26,27] has been developed extensively during the last 10 years, although the seminal works on solar cells date back to the early 1980s by, for example, Eli Yablonovitch [28]. Recent works have shown that a reliable solution to enhance the responsivity of Si-based photodiodes with a thin absorption region can be found by incorporation of two-dimensional (2D) periodic nanoholes into Si surfaces [29,30]. Due to the 2D regular modulation of the refractive indices, the hole array acts like a conventional grating coupler and enables light trapping by conversion of an initial incident vertical plane wave to the lateral collective modes [29,30,31,32]. Propagation of photons in the larger lateral dimension elongates the effective absorption length, benefiting the enhanced optical absorption. Micro- and nanohole arrays have already demonstrated their capability in improving light detection from the visible to the IR spectral range in Si [29,33] and Ge-on-Si [30] surface-illuminated high-speed photodiodes. An increase from 16% to 60% absorption efficiency was demonstrated at 850 nm in avalanche Si photodiodes [34]. A four-fold enhancement of photoresponse was achieved at 2 μm in GeSn/Ge multiple-quantum-well detectors [35]. The responsivity enhancement factors of 1.12 and 1.33 for the near- and mid-IR were observed in a InAsSb-GaSb heterostructure [36]. In our recent paper [37], we observed that incorporation of layers of Ge/Si quantum dots in a two-dimensional photonic crystal with period of 650 nm leads to a five-fold increase in the photocurrent at a wavelength of 1.3 μm. In the present study, we examined hole arrays with different hole diameters and array periodicity as a means to enhance the near-IR photoresponse of Ge/Si QDIPs. We found that QDIPs integrated with photon-trapping holes exhibit a dramatic increase in photocurrent compared to control devices. For a hole array periodicity of 1700 nm and hole diameter of 1130 nm, the responsivity of the photon-trapping device is found to be enhanced by about 25 times at 1.2 μm and by 34 times at 1.6 μm relative to a bare detector without holes.

## 2. Materials and Methods

The samples were grown by solid source molecular beam epitaxy (MBE) using a Riber SIVA-21 system. In the experiments, a (001)-oriented silicon-on-insulator (SOI) wafer with a 100 nm top silicon film and 400 nm buried silicon oxide was used as the substrate. The sample geometry is shown in Figure 1a. Employment of an SOI substrate makes it possible for the guided waves to be confined by the SOI layer due to the large contrast in the refractive index between SiO_2_ (*n*∼1.45) and Si (*n*∼3.42) layers [38,39]. The preparation started with a 20-nm-thick undoped Si layer. Then the heavily doped ($7\times 10^{18}$ cm${}^{-3}$) p-type Si bottom layer with the 200 nm thickness was grown at 600 °C. The active region of QDIPs was composed of ten stacks of Ge quantum dots separated by 10-nm Si barriers and was sandwiched in between the 100-nm-thick intrinsic buffer and 30-nm-thick cap Si layers fabricated at 500 °C and 400 °C, respectively. A two-step process was used to fabricate Si barriers, which included a 1-nm-thick Si layer grown at 250 °C followed by a deposition of a 9-nm-thick Si layer at 400 °C. This procedure allows reduction of Ge-Si intermixing and preservation of the island shape and size from the effect of further high-temperature deposition. Each Ge QD layer consisted of a nominal Ge thickness of about 0.9 nm and formed at 250 °C with a rate of 0.04 nm/s by self-assembling in the Stranski–Krastanov growth mode. The lowest growth temperature is necessary to generate small Ge QDs with high density and abrupt interfaces due to the low adatom surface mobilities. QDs have a hut-like shape with a lateral size of 9.4±3.2 nm and a height of about 1 nm. The surface density of QDs is $5.2\times 10^{11}$ cm${}^{-2}$. From the photoluminescence study, we observed that the ground state interband transitions occur around 1.6 μm [2]. Finally, an Sb-doped 25-nm-thick n-Si top contact layer (∼$10^{19}$ cm${}^{-3}$) was grown at 400 °C to form a pin diode structure. Details of the structure growth can be found in [2,40].

After the MBE growth, the wafers were processed into 700 μm diameter vertical pin QDIPs (Figure 1b). The hole arrays were etched (225 nm deep) up to the bottom p-Si layer with a cylindrical shape profile, serving as the photon-trapping structures. The air holes were produced by means of the reactive ion etching (RIE) of SiGe layers through a metallic template. The template was a 30-nm-thick perforated Cr film formed on the surface of the heterostructure by electron-beam lithography, deposition of the metal in a vacuum, and the subsequent lift-off process. Etching was performed in a CF4 plasma on a Plasmalab System 80 setup in the cyclic etching/cooling regime. The hole arrays have square lattice symmetry with different hole diameter *d* and period *p* values (Figure 1c–e). Devices with d/p=430/650 nm, 930/1400 nm, and 1130/1700 nm were studied. Selected array parameters are characteristic of those chosen in the literature to improve light detection at telecom wavelengths in bulk Ge/Si [30] and GeSn/Si [35] layers. The air volume fraction $f=\left(\pi /4\right){\left(d/p\right)}^{2}≃0.34$ was the same for all devices. A reference QDIP without any etched structures was also fabricated for the comparison of device performance. Both flat and nanostructured samples were taken from a single die of the same wafer right next to each other. Top and bottom ohmic contacts were made using Au evaporated onto the sample surface and annealing at 350 °C for 5 min in an Ar atmosphere (Figure 1b). A 5-nm-thick Ti film was deposited between the 50-nm-thick gold layer and the QDIP to promote the adhesion of the Au film to the Si surface. The samples were glued with silver epoxy onto a ceramic chip carrier. The contacts were bonded with silver wires using thermocompression bonding.

The photocurrent measurements were performed at room temperature. The incident IR light illuminated detectors from their substrate side. This allows the hybridization of the focal plane arrays with a silicon readout integrated circuit on the top of the devices [41]. The normal-incidence photoresponse was obtained using a Bruker Vertex 70 Fourier spectrometer with a spectral resolution of 30 cm${}^{-1}$ along with a SR570 low-noise current preamplifier. The photocurrent spectra were calibrated with a deuterated L-alanine doped triglycine sulfate detector. The room-temperature dark current was tested as a function of bias voltage between −2 V and +2 V by a Keithley 6430 Sub-Femtoamp Remote SourceMeter.

## 3. Results and Discussion

### 3.1. Simulation Results

To demonstrate formation of in-plane quasi-guided modes, we inspected how the actual electric fields look in the structures under study. Calculations of near-field components distribution were carried out using the 3D finite-element frequency-domain (FEFD) method [42] with commercial software Comsol. Floquet periodic boundary conditions were used along the planar *x* and *y* directions to simulate an infinite array of unit cells. Perfectly matched layers were used along the vertical direction (*z* direction) to prevent the reflection of the waves from the top and bottom domain boundaries. The plane wave radiation with a linear polarization along the *x*-axis was incident from the substrate side of the QDIP. The simulated structures are the hole arrays shown in Figure 1c–e. The hole arrays have a square lattice symmetry with a hole depth of 255 nm. The air, SiO_2_, and Si regions were modeled using rectangular parallelepiped geometry with correspondent refractive indices. FEFD numerical simulation of the near-field intensity and the Poynting vector in-plane distribution at λ=1.6 μm are shown in Figure 2 for a sample with a nanostructured surface. Here, the hole array parameters are d=930 nm and p=1400 nm. The other samples behave in a similar way. In a QDIP with photon-trapping holes, the energy flux leaks from the holes into the Ge/Si layers, resulting in formation of guided lateral modes that are concentrated in the semiconductor and could be entirely absorbed by QDs. Since the lateral dimension of the devices is much larger than the thickness of the QD layer, QDIPs with photon traps can be more efficient.

### 3.2. Experimental Results

Figure 3 shows the dark current density-voltage characteristics of pin QDIPs with and without photon-trapping structure. The high on/off ratio of the devices around $10^{3}$–$10^{4}$ is observed, manifesting good rectifying behavior. For the device with a flat surface, the dark current is 0.13 mA/cm^2^ at a reverse bias of 1 V. This value is two orders of magnitude lower than previously reported ones for Ge-on-Si [30,43,44] and GeSn/Ge [45] photodiodes with dislocation-rich Ge active layers. The ultralow dark current density in Ge/Si QDIPs is attributed to the formation of dislocation-free Ge QDs by strain-driven self-organization through the Stranski–Krastanov growth mode. As shown in Figure 3, the current of Ge/Si QDIPs with holes increased by around 3 times compared to the non-photon-trapping counterpart. This is probably due to the formation of crystalline defects and dangling bonds at the hole surfaces during the RIE process, which can contribute to increased current level.

Figure 4 presents the measured responsivity spectra of the QDIPs with and without photon-trapping holes with different reverse bias voltages. The reference device of this figure has a broad response and covers the near-IR spectral range up to 1.8 μm, coming from the interband transitions between the electron states in the conduction band and the hole states bound inside Ge QDs. The weak dependence of the photocurrent on the bias indicates efficient collection of photogenerated carriers. The spectra of the photon-trapping devices consist of a smoothly varying background superimposed on a set of multiple photocurrent peaks, providing evidence for the excitation of an ensemble of guided collective modes [25,46].

The photocurrent enhancement at the zero bias condition is plotted in Figure 5. To obtain the hole-induced photoresponse enhancement factor, the experimental responsivity spectra were normalized to the reference spectrum of the bare sample without the photon-trapping structure. Compared with the conventional flat QDIP, all nanostructured devices provide significant responsivity improvement at the wavelength range from 1.0 to about 1.8 μm. For p=650 nm and d=430 nm, the enhancement factor is about 10. This value is about twice as large as that previously reported in [37] due to the thicker SiO_2_ layer (400 nm) in the present devices. The enhancement factor rises with the increase in the hole array parameters *d* and *p* due to the increase in the optical mode density [25]. For the hole array periodicity of 1700 nm and hole diameter of 1130 nm, the responsivity of the photon-trapping device is found to be enhanced by about 25 times at 1.2 μm and by 34 times at 1.6 μm relative to a bare detector without holes.

It should be noted that microhole array absorption enhancement can occur not only as a result of guided modes, but also due to antireflection, resulting from the reduction of the refractive index difference between air and the device layers [25,30,35,47]. However, since the typical reflection losses of a bare device are about 30%, the antireflection property cannot lead to a multiple absorption enhancement.

## 4. Conclusions

We would like to draw attention to the fact that the sort of photonic crystal implemented in our devices can be impractical as a light-trapping structure for solar cells because of the large number of undesired defects introduced across the device, which degrade the cell performance [26]. In contrast, using QDs as light absorbers, the most photogenerated electron–hole pairs are isolated from the hole surfaces, and therefore the photodetector might still outperform the bare case. Of course, some losses will happen in vertical transport. However, the benefit from the increased absorption coefficient can exceed the carrier losses on the hole surfaces, giving rise to the responsivity improvement.

In summary, we have reported on vertical Ge/Si QD pin photodiodes with self-assembled Ge quantum dots grown on an SOI substrate for photodetection in the near-infrared telecommunication wavelength range. The photon-trapping 2D hole arrays were introduced into QDIPs to convert the incident electromagnetic radiation to the lateral collective modes. The periodic hole arrays have square lattice symmetry with different hole diameter *d* and period *p* values. Devices with d/p=430/650 nm, 930/1400 nm, and 1130/1700 nm were studied. Compared with the non-photon-trapping counterpart, the photon-trapping QDIP exhibits a 25× responsivity enhancement at a wavelength of 1.2 μm and a 34× enhancement at 1.6 μm for the hole array periodicity of 1700 nm and hole diameter of 1130 nm. Additional functionality can be achieved by adjusting the position and number of the QD layers, array periodicity, size, and the depth of the holes.

## Figures and Tables

**Figure 1 nanomaterials-11-02302-f001:**
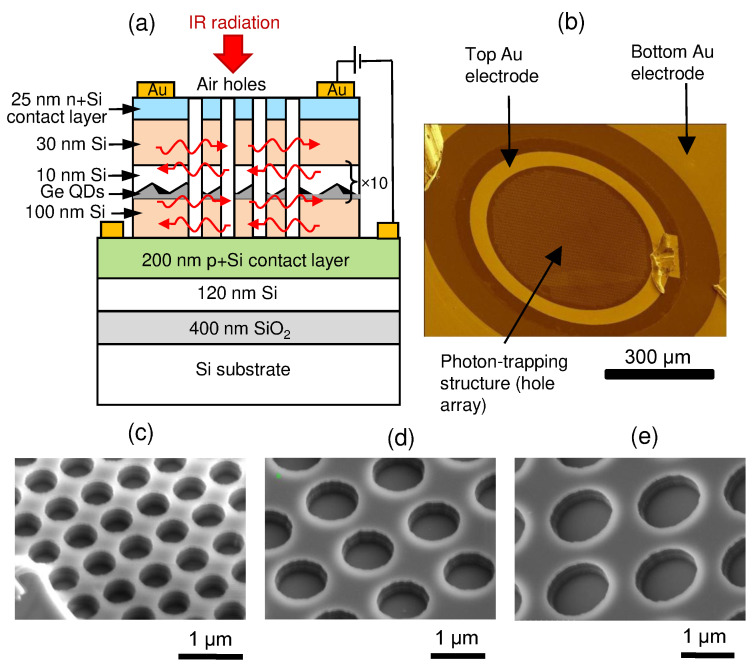
(**a**) PIN QDIP structure on an SOI wafer showing the integrated array of holes (photon-trapping structure) that spans the n- and i-layers. (**b**) Plane view scanning electron microscopy (SEM) image of the Ge/Si QDIP with photon-trapping structure. (**c**–**e**) Zoom-in SEM micrographs of the hole arrays with different period *p* and diameter *d*. (**c**) *d*/*p* = 430/650 nm, (**d**) 930/1400 nm, and (**e**) 1130/1700 nm.

**Figure 2 nanomaterials-11-02302-f002:**
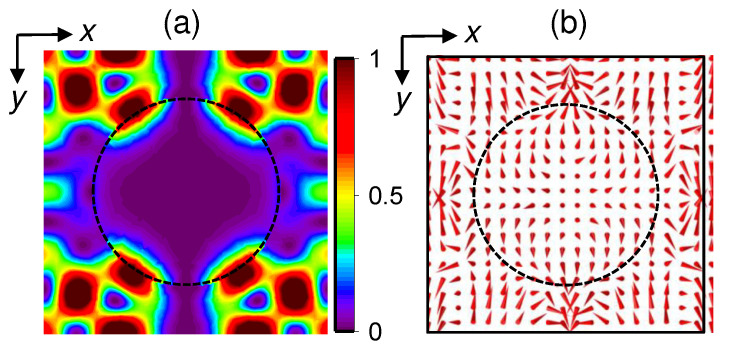
Lateral distribution maps of (**a**) the electric field intensity and (**b**) the Poynting vector in the (x,y) plane in photodetectors with nanostructured surfaces. The hole array parameters are d=930 nm and p=1400 nm. The data were taken at the distance of 105 nm below the device surface. The boundary of the hole is drawn by the dashed line. Arrows indicate the directions of the Poynting vector. The structure was illuminated from the substrate side with a plane-wave light at 1.6 μm polarized in the *x* direction. The simulated region is a unit cell of hole-array structure. Excitation of the guided lateral modes concentrated outside the hole is evident.

**Figure 3 nanomaterials-11-02302-f003:**
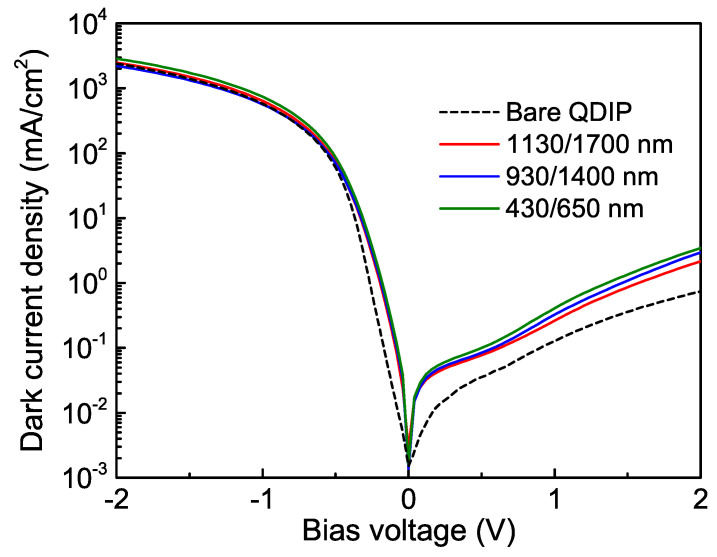
The biasdependence of the dark current density in the Ge/Si QDIPs with and without photon-trapping hole arrays. The hole diameter/period values are 430/650 nm, 930/1400 nm, and 1130/1700 nm.

**Figure 4 nanomaterials-11-02302-f004:**
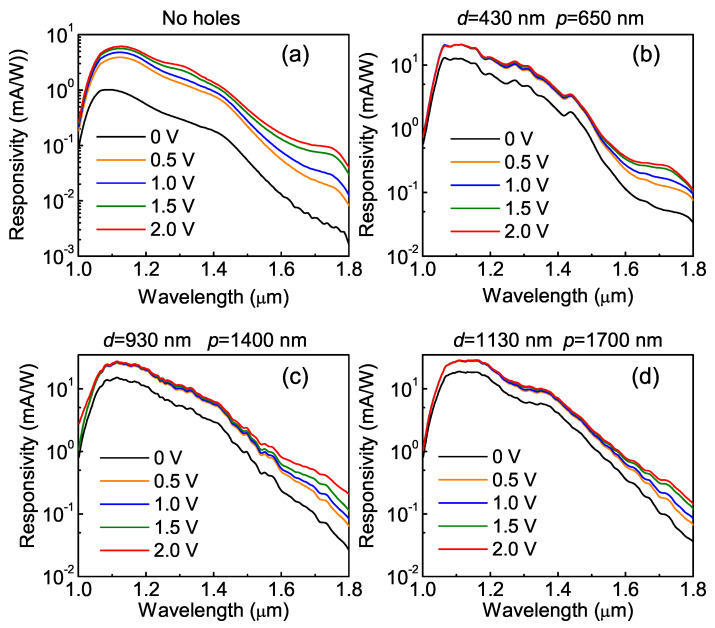
Responsivity spectra of the Ge/Si QDIPs (**a**) without and (**b**–**d**) with photon-trapping hole arrays. The hole diameter/period values are (**b**) 430/650 nm, (**c**) 930/1400 nm, and (**d**) 1130/1700 nm.

**Figure 5 nanomaterials-11-02302-f005:**
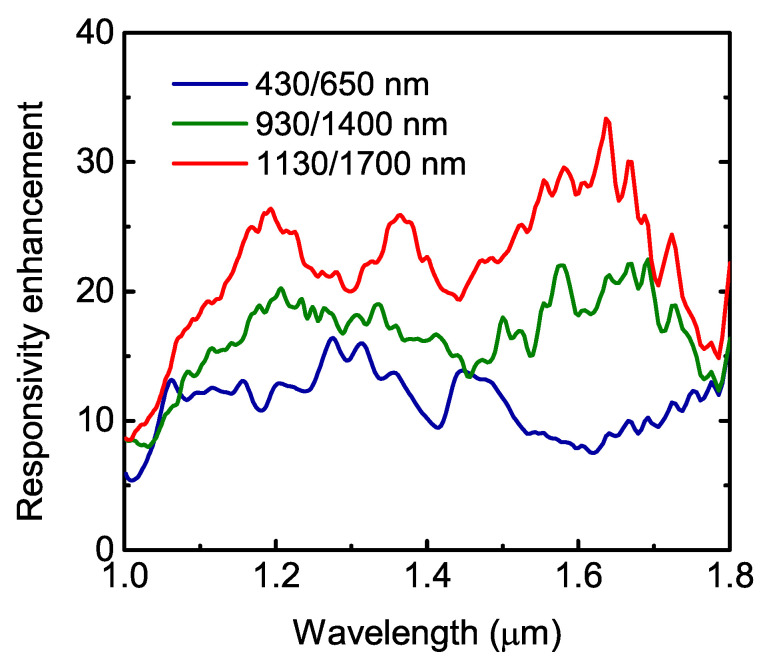
Responsivity enhancement spectra of the Ge/Si QDIPs with photon-trapping hole arrays.

## Data Availability

Data are contained within the article.

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
