# Peer review of "Near-Infrared Photoresponse in Ge/Si Quantum Dots Enhanced by Photon-Trapping Hole Arrays"

_nanomaterials, 2021, doi:10.3390/nano11092302_

Round 1

Reviewer 1 Report

The authors present an experimental realization of a light-trapping structure to enhance the performance of QD-based IR photodetectors. My general comment is that the experimental effort is unfortunately not reflected in the quality of the manuscript. I found the idea and approach not very original, although its physical realization worths recognition. My recommendation is to implement a major revision of the manuscript addressing the following comments:

  1. The proposal of using a light-trapping structure to enhance the absorption or emission of optoelectronics devices such as LEDs, solar cells, or photodetectors is not new and has been developed extensively during the last 10 years, although the seminal works on solar cells date back to the early '80s by, for example, Eli Yablonovitch. I am perfectly aware that the current manuscript is not a review. However, the introduction fails in framing the correct context of the research. At the moment the authors only focus on photodetectors and by adding a few lines the visibility of the study could be improved.
  2. In line with this, one detail that the authors have overlooked, and I think that they should highlight, is the fact that the sort of photonic crystal (PC) implemented in their device is impractical as a light-trapping structure for solar cells because of the large number of defects introduced across the device. In contrast, using QDs as light absorbers, the photogenerated e-h pairs are isolated from the PC hole surfaces, and therefore the photodetector might still outperform the bare case. Of course, some losses will happen in vertical transport. It would be interesting if the authors could add a comment in this line. Why do air holes in solar cells reduce the efficiency while in photodetectors increase the responsivity?
  3. I have a few comments on the section "Simulated Results". First, I would suggest the title "Simulation Results" as simulated could be ambiguous. Second, the simulation only shows qualitative results which could be otherwise referenced from the literature. The main conclusion out of this section is that the PC has in-plane quasi-guided modes and that coupling is possible. These, for example, are well reported in [1]. Moreover, Figure 2 (b) is not really informative as it is well-known that in a multilayer stack there is no diffraction and light propagation can be described by forward and backward modes, hence, no out-of-plane is possible without in-plane texturing. In third place, I would have expected in this section some sort of simulation showing why the authors have chosen the values of p and d for the fabrication. After reading the manuscript it is unclear what is the rationale behind it. I could have overlooked it, although the right place to discuss the optimal or close to optimal values of d and p is this section. 
  4. My final comment concerns that claim on the outperformance of the PC photodetector. The authors compare a bare semiconductor device (with typically 30% reflection losses due to the high refractive index contrast between semiconductors and air) with a PC device. A fair comparison would be between an anti-reflective coating device vs. a PC device. There is no discussion on the role of the reflection losses between the bare and PC devices, which I think is critical to support the main claims appearing in the abstract. I can perfectly understand that asking for new devices plus measurements is too much work. But adding such an analysis in Section 3.1 is possible. 

[1] S. G. Tikhodeev, A. L. Yablonskii, E. A. Muljarov, N. A. Gippius, and T. Ishihara, “Quasiguided modes and optical properties of photonic crystal slabs,” Phys. Rev. B, vol. 66, p. 045102, (2002), DOI:10.1103/PhysRevB.66.045102.

Reviewer 2 Report

This is a good paper, it is cleary written and the results and the discussion are convincing. The scientific novelty is moderate, because the authors recently published the basic idea in JETP letters 113, 498 (2021). However, they present some new models and data (e.g. impact of of hole period on responsivity, modification of substrate), and therefore a publication is justified, however, they should cite their paper as a reference.

Author Response

We cite our previous paper as Ref. 37.

Round 2

Reviewer 1 Report

The current version of the manuscript reflects that the authors have addressed many of my comments. I was expecting a different approach in some of them, but the changes are enough to support the publication of the manuscript in Nanomaterials.